# Mental Fatigue after Mild Traumatic Brain Injury in Relation to Cognitive Tests and Brain Imaging Methods

**DOI:** 10.3390/ijerph18115955

**Published:** 2021-06-02

**Authors:** Birgitta Johansson

**Affiliations:** Institute of Neuroscience and Physiology, The Sahlgrenska Academy, University of Gothenburg, 413 45 Göteborg, Sweden; birgitta.johansson@neuro.gu.se

**Keywords:** mild TBI, concussion, mental fatigue, cognition brain imaging

## Abstract

Most people recover within months after a mild traumatic brain injury (TBI) or concussion, but some will suffer from long-term fatigue with a reduced quality of life and the inability to maintain their employment status or education. For many people, mental fatigue is one of the most distressing and long-lasting symptoms following an mTBI. No efficient treatment options can be offered. The best method for measuring fatigue today is with fatigue self-assessment scales, there being no objective clinical tests available for mental fatigue. The aim here is to provide a narrative review and identify fatigue in relation to cognitive tests and brain imaging methods. Suggestions for future research are presented.

## 1. Introduction

Traumatic brain injury (TBI) is a common health problem and can lead to temporary or permanent disabilities of cognitive or physical functioning [1,2,3], and approximately 70–90% of injuries are mild TBI (mTBI) [4]. Most people recover after an mTBI within one to three months, but those who suffer long-term symptoms experience a reduced quality of life with deterioration of general health, cognitive impairments, higher risk of depression, social isolation, behavioral problems, and an inability to maintain employment status or continue with education [5,6,7,8]. Thus, “mild” injury may be anything but mild [8].

After a traffic-related mTBI, 23% did not recover within a year [9]. An unfavorable outcome was reported for 30% of all mTBI patients [10]. A review reported that approximately half of the individuals with a single mTBI demonstrated long-term cognitive impairment [11]. A worse outcome can develop after recurrence of mTBI [12]. The assessment and definition of symptoms after an mTBI varies between studies, and symptoms are also common in the population irrespective of whether it is a case of an mTBI. The interaction between pre- and post-biological and psychological factors results in a complex model [13,14], and there is limited evidence for variables as predictors of fatigue [14]. However, in many studies, mental fatigue is not recognized as a separate factor, despite being one of the most distressing and long-lasting symptoms following mTBI [15,16], having an impact on well-being and quality of life [2]. Further research is warranted to better understand the associations between fatigue and pre- and post-morbid variables and for the development of adapted treatment options.

Pathological mental fatigue (not to be confused with non-pathological mental fatigue) is a reduced capacity to continue a mental activity repeatedly. It is a lack of mental energy and a reduced ability to restore the energy in response to performed activity. The mental energy varies depending on the activities performed, but it never returns to normal and is not restored after sleep or an extended rest [17]. Mental fatigue restricts the ability to perform daily activities with work, studies, and social activities, and there is a limited endurance over the course of the day and for weeks and months. The mental fatigue is unlike that normally experienced; rather, it persists as a profound exhaustion over months or years, or it is sometimes lifelong. People appear normal, but the fatigue they describe goes far beyond and far deeper than a person without brain fatigue could imagine.

The purpose here is to provide a narrative review of the role of mental fatigue in relation to cognitive tests and brain imaging methods after TBI. Both terms, concussion and mTBI, are used.

### 1.1. TBI and Prevalence of Long-Term Fatigue

The precise estimation of fatigue after mTBI depends on the assessments used. Nonetheless, fatigue is often reported as having a bearing on everyday life. One third of patients who had suffered an mTBI reported fatigue at 6 months as well as a decrease in physical and social activities [18]. A consistently high frequency of fatigue after TBI was found during the first six months after the TBI [19], and 40% suffered from headache and fatigue one year after mTBI [20]. After 5 years, 73% were still suffering from fatigue after a TBI, affecting their everyday lives [3]. After ten years, the fatigue was still present, irrespective of injury severity [21]. Improvement was reported during the first year, after which time it was limited [22].

### 1.2. Origin of Fatigue after TBI

The origin of fatigue is not fully understood and is often suggested to be related to multifactorial factors including biological, psychological, and social aspects. Fatigue is thus a concept with a subjective feeling of exhaustion possibly having a biological as well as a psychological origin and can be difficult to manage depending on the adapted strategies used and social factors involved. However, fatigue in neurological disorders is suggested to be related to circuits that connect the basal ganglia, amygdala, thalamus, and frontal cortex [23]. This is supported by brain imaging studies with TBI participants [24,25,26,27,28,29]. The integration of these areas is important for appropriate behavior and cognitive functioning [30] and also for executive function such as motivation, learning, planning, goal-directed behavior, and emotion regulation [31]. Genetic factors also interact with TBI, and carriers with the APOE epsilon4 allele reported fatigue twice as often after compared with before an mTBI, while no increase in fatigue was found in a non-injured control group [32]. A long-term follow up 1–5 years after a TBI showed a carrier with the APOE epsilon4 allele reporting a lower rating on the Glasgow Coma Scale—Extended (GOSE) [33]. In the study by Ponsford et al., fatigue was not measured, but reduced social and leisure activities from GOSE were related to fatigue after TBI and mTBI [34], and fatigue may have been present among the TBI participants [33].

TBI has a heterogeneous pathology. External mechanical or rotational forces result in focal injuries due to contusion, laceration, intra- and/or extra cerebral haemorrhage, and diffuse brain damage. Metabolic and structural lesions that disrupt the pathways interconnecting the basal ganglia, thalamus, limbic system, and higher cortical centre are suggested to be involved in the pathophysiological process of fatigue in neurological disorders [23]. In multiple sclerosis, fatigue is hypothesized to be related to impaired dopamine communication between the striatum and prefrontal cortex [35].

The relation between mental fatigue and a secondary injury that gives rise to a neurochemical cascade that can induce inflammation, biochemical imbalance, oxidative stress, and extended reduction in glucose metabolism [36] is not well understood. However, a theoretical model for the development of mental fatigue in relation to neuroinflammation and disturbed neurochemical balance is presented [37]. Brain injury in the acute phase causes a neuroinflammation, with microglia and astrocytes forming inflammatory substances, protecting the nerve cells from damage. The cytokines (e.g., Tumor Necrosis Factor-alpha and Interleukin 1-beta) initially attenuate the signaling between the brain cells, which is important in the acute phase when energy is needed to repair the nervous system. If such substances persist after the injury, they may instead lead to reduced efficiency in nerve cell traffic, resulting in impaired cognitive function and perception of mental fatigue [37,38].

### 1.3. Fatigue Self-Assessment Scales

The best method measuring fatigue today is self-assessment scales. Several scales exist, although there is no consensus regarding the assessment scales. Fatigue can be divided into state- or trait-fatigue. State-fatigue or fatigability is measured as a reduced performance or a worse perception of fatigue in relation to a specific activity performed. Visual analog scales (VAS) are commonly used for fatigability, but also cognitive tests or brain activation are used. Trait-fatigue is measured with assessment scales as the subjective perception of fatigue over a pre-defined period of time (weeks and months) [39]. In relation to the mental fatigue discussed here, both state- and trait-fatigue are valid. The mTBI individual suffering from long-term or chronic mental fatigue will have a long-lasting fatigue and will not be fully recovered, and the mental fatigue will vary over the course of the day and from one day or week to the next. When suffering from long-lasting mental fatigue, there is also a distinct fatigability in relation to performed mental activities, and it takes a long time to recover from the extra load on the brain. Physical activity has not been extensively investigated in terms of mental fatigue and mTBI, but patients commonly report a worsening of their mental fatigue following physical activity.

### 1.4. Emotional Distress and Fatigue

Emotional and psychological problems before the brain injury as well as stress add to the trauma, and personal coping abilities interact with psychological, biological, and genetic factors [13,40]. These need to be taken into account for a multifactorial interactive model for mental fatigue during clinical assessment. Clearly, factors such as depression or unemployment before the injury will persist during recovery. Long-lasting fatigue after an mTBI will affect everyday life, such as not managing work and social activities. This can result in an increase in psychological distress if adequate rehabilitation or helpful personal coping strategies are not in place. People suffering from depression and anxiety prior to a skull trauma more frequently experience long-term complications and psychiatric disorders after a brain injury [41,42,43]. With regard to executive function, self-report of executive function increased after mild to moderate TBI and was found to be a predictor for depressive symptoms, although this was not related to objective tests for executive function [44]. Fatigue was not measured in the study. Many authors have reported depression to be common after a brain injury, but they have seldom included fatigue in their analyses.

According to Cantor et al. [16], overlapping symptoms between fatigue and depression can give misleading results if the core symptoms are not carefully evaluated. When considering fatigue after a TBI, it is suggested to be related to the brain injury itself [2] rather than as a consequence of anxiety and depression [45], even though emotional distress can aggravate the symptoms of fatigue. Fatigue was also found to uniquely contribute to disability after TBI, measured with the Mayo–Portland Adaptability Inventory, after controlling for injury severity, executive functions, and depression status [46]. Fatigue decreased during the first year after mTBI and depression, insomnia, cognitive difficulties, and work status were related to the fatigue [47]. Adjustment for the overlapping items in the Mental Fatigue Scale (MFS) and Comprehensive Psychopathological Rating Scale (CPRS—depression and anxiety) removed the differences in depression and anxiety between participants with an acquired brain injury (mTBI, stroke) and healthy controls, but not for the mental fatigue [48]. It is important to acknowledge the increased risk for depression and suicide after a TBI [49] and also to understand the significance of this within the context of fatigue. Mental fatigue cannot be explained within the context of emotional distress alone. However, treatments for depression and anxiety are essential, and these can reduce mental fatigue and improve well-being [50,51].

### 1.5. Cognitive Function in Relation to Fatigue

From the book by Lezak et al., p 177 [52], the effort to accomplish ordinary activities after a brain injury is exemplified.


*“Activities that are normally automatic but become effortful after the injury include many that are performed frequently throughout a normal day, such as concentrating, warding off distractions, reading for meaning, doing mental calculations, monitoring ongoing performance, planning the day’s activities, attending to conversations at once, or conversing with background noise, etc. It is by little wonder that by late afternoon, if not by noon, many of these patients are exhausted”.*


Despite the awareness of these problems, it has been difficult to detect such cognitive complaints after mTBI using standard neuropsychological tests, with the result that cognitive difficulties can go undetected [53,54,55]. In the clinic, many patients who suffer from long-term difficulties after an mTBI may perform within a normal range on cognitive tests, although they may still report mental fatigue, including distractibility and difficulties with reading as well as executive function. Executive function is a broad over-reaching term for cognitive and emotional functions and is suggested to have one of the greatest effects on functional outcomes after a TBI [56], irrespective of TBI severity, having a negative impact on daily living and the ability to go back to work and studies [31]. Executive function is important for the ability to respond adaptively to novel situations and manage emotional and social skills, including verbal reasoning, problem solving, attention control, sustained attention, planning, judgment, decision-making, motivation, impulsivity, resistance to interference, utilization of feedback, multi-tasking, and cognitive flexibility [52,54,55]. From a large-scale study including mild to severe TBI subjects, information processing speed was identified as the domain most strongly associated with daily functioning [57]. Neuropsychological tests have been developed to identify focal and diffuse brain damage rather than to predict how a patient would function in his or her environment [58].

The results reported here include both mTBI and TBI patients, the subject of few studies (Table 1). Fatigue in a group with mild to severe TBI was related to the speed subscales included in the Cambridge Neuropsychological Test Automated Battery (CANTAB) [59]. A worsening of performance after severe TBI on a dual-task with load on working memory and/or executive control was related to increased mental effort. Mental effort was not related to task difficulty or to performance, and it was suggested that fatigue interferes with performance even in simple and automatic tasks [60]. Patients with severe TBI who have a high baseline fatigue performed less well on the selective attention test with a longer reaction time and more omissions compared to controls. Significant correlations were found in the TBI group between attention performance, mental effort, and subjective fatigue. Depression did not correlate with fatigue [61]. Reduced processing speed (digit symbol-coding, reading speed, trail making test) correlated with mTBI and TBI participants all suffering from mental fatigue. Those participants with mTBI working full-time rated their mental fatigue on a similar high level as participants with mTBI and TBI on sick leave. Unlike in the case of those on sick leave, objective cognitive tests did not differ between controls and those with mTBI who were also working full-time. This was suggested to be related to the mental demands during the day. Those suffering from mental fatigue after an mTBI but who are also working full-time might need to devote more time and effort to mental work [34]. Slower performance on a complex selective attention task was associated with fatigue for mild to severe TBI participants in comparison with controls. It was suggested that tasks with a higher order of attentional demand in everyday life are associated with subjective fatigue, this being irrespective of the effects of mood [62]. In contrast, a group including TBI participants reported increased fatigue after the test session having a slower reaction time compared to controls, although the reaction time remained constant for both groups during the session [25]. A vigilance test with a total duration of approximately 45 min was performed by participants with mild to severe TBI. The TBI group was slower and remained at a similar slower speed throughout the duration of the task. Higher state-fatigue was associated with more omissions for TBI participants. A subgroup of these participants showed a decline in performance, which the authors suggested was associated with the greater psychophysiological cost of striving to maintain a stable performance over time, and this cost was associated with a reported increase in fatigue [63]. Reaction time was longer and increased during a 20-min vigilance task for mTBI group, while the reaction time remained stable for the controls [26]. Decreased performance (fatigability) was reported for an mTBI group who described more fatigue than controls in more demanding cognitive tests with simultaneous activation of several cognitive domains (Digit Symbol Substitution Test—DSST) and executive functions (Color Word Test). However, fatigability was not found to be present for less demanding tests involving simple automatic attention [64]. Using the Barrow Neurological Institute (BNI) Fatigue Scale and BNI Screen for Higher Cerebral Functions, the TBI group reported significantly higher levels of fatigue compared to controls. Fatigue was unrelated to injury severity, number of days since injury, cognitive impairment, and gender. The authors suggested regular rest during the day for inpatients in rehabilitation units [65]. During a subacute period, no differences between mTBI and controls in simple reaction time tasks was found, but with increased task complexity, the mTBI group performed more slowly than controls. However, no relation between cognition and fatigue was found [66]. Fatigue and relations to saccadic eye movements and attention in mTBI patients and patients with minor orthopedic injuries early after an mTBI was investigated. The mTBI group reported higher self-rated change in fatigue compared with levels prior to the mTBI and showed more fatigability/state-fatigue measured with cognitive tests. Their state-fatigue correlated with prosaccade latency and cognitive fatigability. In contrast, mTBI trait-fatigue correlated with anxiety and antisaccade latency and variability [67]. 

When combining endurance and cognitive function among people suffering from mental fatigue after a TBI, no change was found when tests were repeated, while improvement was achieved for healthy controls [24,59,68,69]. A 30-min computerized neuropsychological test battery using the Cambridge Neuropsychological Test Automated Battery with three repetitions, with the second and third repetition separated by approximately 2 h of interviews and self-report measures, was compared between TBI and control groups. Fatigue was associated with poor performance in speed for the TBI group, but this was not associated with an objective decline in cognitive tests. On the contrary, improvement on speed and accuracy was found in the control group [59]. A five-minute computer test with five repetitions, including task repetitions with simultaneous load on processing speed and working memory, showed an overall slower speed for the mTBI group. During the test, the controls became significantly faster when they repeated the task, while this was not found for the mTBI group, who remained on a similar slower speed [69]. After a 2.5-h test session, there was no change in test performance for the mTBI group (processing speed, working memory, attention), but a significantly reduced mental energy compared to the controls was reported after the test session. In contrast, the control group improved their processing speed (Digit symbol coding, WAIS-IV) when the test was repeated [24]. Repetition (three times) of a working memory and wordlist memory test, with 60 sec. rest between trials, revealed no difference between control and mTBI groups after the first trial. However, a significant difference was reported between groups in working memory after trial three, and a significant difference was also found between groups for word-list learning after trials two and three, with the controls performing better with a greater improvement in word-list learning. It was suggested that repetition of tests can be useful for distinguishing cognitive fatigue [70].

In summary, several studies have demonstrated that fatigue is related to information processing speed, attention, and memory. More demanding cognitive tests are suggested to be related to fatigue, but also automatic and simple cognitive tests are suggested to be related to fatigue. This is not a simple relationship, and results can vary depending on which tests and study design are used. However, in no case has improved cognitive function been reported for those suffering from fatigue after an mTBI or TBI. The measurement of endurance, using the repetition of tests, is a promising means of assessment. Fatigability, namely a worsening of performance, is not an accurate means of assessment. The preferred method of assessment is altered performance over time or after repetition of tasks compared to controls.

It can be argued whether the injury per se is related to impaired cognitive function and, to a lesser extent, to mental fatigue. However, from a study including a recovered mTBI group, no difference was found in mental fatigue and reaction time between those who had recovered and the controls, while a significantly higher MFS and a slower reaction time were reported for those with persistent mTBI [71].

### 1.6. Brain Imaging in Relation to TBI and Fatigue

Brain imaging as an assessment tool for fatigue after TBI is in its infancy. A few recent studies have shown promising results (Table 2). For participants with a moderate to severe TBI, increased brain activity was reported during a one-hour scanning, while controls showed decreased brain activity in a processing speed task, Symbol Digit Modalities Test —(SDMT), measured with functional magnetic resonance imaging (fMRI). Increased brain activity was suggested to represent increased effort and fatigue. No assessment of trait- or state-fatigue was performed [27]. In contrast, when the same processing speed test (SDMT) was performed, lower brain activity (fMRI) in the basal ganglia, primarily the caudate nucleus, thalamus, and anterior insula, was reported for the TBI group (ranging from mild to severe) compared to controls. The brain activity decreased across the 27-min test session for the controls, whereas the TBI group remained on a similar lower activity level and reported an increased state-fatigue after the session compared to controls [25]. A group of moderate to severe TBI patients reported increased fatigue during the test session compared to controls. Brain activation showed a positive relation between fatigue and fMRI for the more advanced 2-back working memory task and a negative relation for the simpler task (0-back task). Fatigue in relation to brain activation was related to the caudate nucleus [28]. Using functional near-infrared spectroscopy (fNIRS) to measure brain activity during a 2.5 h session, a lower event-related oxygenated haemoglobin (oxy-Hb) concentration was detected in the frontal cortex for the mTBI group, all of whom were suffering from mental fatigue compared to controls. The Stroop–Simon test was repeated once, and other cognitive tests and assessments scales were performed between these. The event activation remained on a similar lower level for the mTBI group and on a similar albeit higher level for the control group. An interaction was found, with the mTBI group having a similar lower oxy-Hb concentration for both congruent and incongruent trials, whereas the controls had a higher concentration of oxy-Hb in the more demanding incongruent trial compared to the congruent trial. Higher mental fatigue correlated with lower oxy-HB [24]. A group suffering from fatigue after mTBI used different brain networks compared to healthy controls during a 20-min vigilance task. There was a correlation between fatigue and functional connectivity in the thalamus and middle frontal cortex and changes in regional cerebral blood flow (rCBF). rCBF was related to fatigability with increased rCBF in the right middle frontal gyrus. Self-rated fatigue was related to increased rCBF in the left medial frontal and anterior cingulate gyri and decreases of rCBF in a frontal/thalamic network [26,29]. To evaluate mental fatigue associated with mTBI in acute and chronic phases, a 20-min psychomotor vigilance test was related to arterial spin labeling–fMRI. Sustained attention was impaired in mTBI patients both in acute and in chronic phases compared to controls, and with worse performance in the acute phase [72]. Brain activation associated with effort and fatigue did not differentiate the mTBI and controls, while functional connectivity did. Connectivity between the left anterior insula, rostral anterior cingulate cortex, and right-sided inferior frontal regions correlated with effort level and state-fatigue in mTBI participants. These connections also correlated with effort level in the controls. Left insula and superior medial frontal gyrus correlated with fatigue. The authors propose a complex link between effort and fatigue and that it may be related to an inefficient neuronal system [73]. Activation assessed with rCBF within the cerebellar cortex correlated with PASAT, measuring information processing and sustained and divided attention in controls, while mTBI participants with self-reported cognitive fatigue showed significant correlations in the inferior frontal and superior temporal cortices [74]. MRI revealed grey matter (GM) and white matter (WM) brain lesions in a TBI group, but fatigue was not found to be related to brain lesions [75]. White matter hyperintensity lesions (WMH) were more common in mild–severe TBI patients compared to controls, but WMH lesions were not related to cognitive tests. Increased WMH lesions correlated with reduced fatigue, but the relation to fatigue was reported to be weak [76]. An mTBI group had a larger number of low diffusion tensor imaging measures (fractional anisotropy values) and more post-concussion symptoms (fatigue not measured separately but included in the PCS) compared to controls. No difference in neurocognitive tests was found [77]. In mild to moderate TBI participants, compared to controls, decreased white matter integrity of the left anterior internal capsule was associated with a greater level of fatigue [78]. A higher rating of fatigue in an mTBI group was associated with decreased right and left thalamic volumes [79]. From the Vietnam Head Injury Study registry, individuals with penetrating brain injuries (PBI) were assessed with a self-report questionnaire and computed tomography (CT) scans. Lesion location and volume were evaluated. PBI patients were divided into three groups according to lesion location: a nonfrontal lesion group, a ventromedial prefrontal cortex (vmPFC) lesion group, and a dorso/lateral prefrontal cortex group. Fatigue scores were compared between the three PBI groups and the healthy controls. Individuals with PBI with vmPFC lesion were significantly more fatigued than the other groups as well as the healthy controls. VmPFC volume correlated with fatigue scores showing that the larger the lesion volume, the higher the fatigue scores [80].

In summary, the brain imaging results are not consistent, although they show altered activity and connectivity in relation to fatigue after TBI. From the fMRI and fNIRS fatigue studies above, experimental block or event-related designs were used. Results need to be understood in relation to the study design used, as the results can vary depending on design. Discordance between study designs was reported for brain tumor patients [81]. No studies relating glucose metabolism (FDG-PET[(18)F]fluorodeoxyglucose—positron emission tomography) or magnetoencephalography (*MEG*) to fatigue in TBI were found, but it would be of interest to more deeply investigate energy metabolism in relation to fatigue after TBI and also to use MEG, a promising method for identifying mTBI [82].

## 2. Suggestions for Future Research

The heterogeneity was high between studies, with insufficient evidence to determine any cognitive or neuroimaging test to predict fatigue after TBI. However, the studies imply connections between cognitive function and brain activation in relation to fatigue after TBI. This is worth exploring further with an intention of being open-minded and considering explanations at many different levels.

Assessment of fatigue must be improved. Here, different fatigue assessment scales were used in the reports. The construction of the scale may have an impact on the results, depending on sensitivity and specificity. It is also important to differentiate between normal and pathological fatigue, as biological and psychological factors may have a different explanation for mental fatigue.

Mental fatigue and its relation to depression needs to be acknowledged after TBI, as it will be important for treatment strategies.

With regard to mental fatigue affecting everyday life, executive and cognitive function have not been explored adequately and it is important to reach a clearer understanding of these aspects in relation to fatigue.

On a group level, slower processing speed is encountered primarily in relation to mental fatigue. More demanding tests may be more sensitive in detecting fatigue, but there is no clear connection to fatigue. Endurance in everyday life is critical for those suffering from mental fatigue, and repetition of tests may be more valid for measuring fatigue.

Brain imaging methods show promising results, establishing an association between fatigue and brain activation, primarily in the frontal cortex, thalamus, caudate nucleus, and connectivity between these areas. Cognitive and physical functions can be related to focal brain injuries. When no single area is clearly affected and cognitive tests are performed within the spectrum of normality and yet debilitating mental fatigue persists, we need to investigate further. We need to better understand the impact of the neurochemical cascade that can induce inflammation, biochemical imbalance, oxidative stress, and an extended reduction of glucose metabolism in relation to mental fatigue [36]. Brain imaging is a growing research field along with other imaging methods such as PET and MEG, which also warrant investigation.

Mental fatigue is not a new phenomenon, and it is essential to acknowledge the debilitating fatigue that patients describe despite the limited availability of diagnostic criteria and the lack of a clear definition of mental fatigue after TBI or mTBI and its origins.

## Figures and Tables

**Table 1 ijerph-18-05955-t001:** Research relating fatigue in adults after TBI to cognitive function. Summary of study characteristics, including details on study sample, methods, and results.

Reference	Number of Participants and Injury Severity	Time Since Injury	Age Year Mean (sd)	Sex (Males/ Females)	Cognitive Test Fatigue Measure	Results
Anderson & Cockle, 2021 [66]	84 mTBI 47 HC	60 (11) days	mTBI 37 (14) HC 34 (10)	mTBI 61/23 HC 18/29	Symbol Digit Modality Test, N-back test and a Increasing Distractors Paradigm assessing reaction time under conditions of increasing cognitive load. MFI	No differences between mTBI and controls on simple tasks, but with increased task complexity, the mTBI group performed more slowly than controls (*p* < 0.05). The difference in cognitive performance was unrelated to fatigue.
Ashman et al., 2008 [59]	202 TBI; 56 mTBI 56 104 moderate/severe TBI 42 unknown 73 HC	At least 12 months after injury, 15 (12) years	TBI 47 (12) HC 41 (12)	TBI 109/93 HC 28/45	Cambridge Neuropsychological Test Automated Battery, repeated 3 times. The second and third administrations of the battery separated by 2 h of interviews and administration of self-report measures. Global Fatigue Index State-fatigue rating on Likert scale at the beginning and end.	TBI group performed worse at all 3 time points (*p* < 0.05). Time by group interaction indicated improved speed between T1 and T2 for controls (*p* = 0.04). TBI group did not vary significantly across all 3 trials. State-fatigue was related to speed subscale at all trial (*p* < 0.05).
Azouvi et al., 2004 [60]	43 severe TBI 42 HC	10 (11) months	TBI 26 (8) HC 27 (9)	32/11 HC matched	Visual-go–no-go task VAS-fatigue	A worsening of performance after severe TBI on a dual-task with load on working memory and/or executive control (interaction *p* > 0.05).
Belmont, Agar, & Azouvi, 2009 [61]	27 severe TBI 26 HC	9 (5) months	TBI 32 (9) HC 32 (10)	TBI 21/6 HC 20/6	Go/No GO (Selective attention task) During the break between the 2 parts of the task (T1), and at the end of the second part of the task. (T2) FSS VAS-fatigue	Patients with severe TBI with a high baseline fatigue performed less well on the selective attention test with a longer reaction time (*p* < 0.001) and more omissions compared to controls (*p* < 0.001). Significant correlations were found in the TBI group between attention performance, mental effort, and subjective fatigue (*p* < 0.05).
Berginström et al., 2018 [25]	57 TBI; 7 sever 10 moderate 40 mild 27 HC	9 (7) years	TBI 42 (13) HC 38 (12)	TBI 31/26 fem HC 14/13	27 min modified SDMT FSS MFS	TBI reported increased fatigue after the test session having a slower reaction time compared to controls (*p* < 0.001). Reaction time remained constant for both groups during the session.
Borgaro, Gierok, Caples, & Kwasnica, 2004 [68]	47 TBI; 18 severe TBI 18 moderate TBI 11 mTBI HC 30	24 (17) days	TBI 36 (16) HC 36 (20)		BNI screening for cognitive function BNI fatigue scale	TBI reported significantly greater levels of fatigue compared to controls. Fatigue was unrelated to injury severity, number of days from injury to assessment, cognitive impairment, and gender.
Johansson et al., 2009 [34]	60 TBI; 14 mTBI fulltime work (FW) 34 mTBI 12 TBI moderate/severe 40 HC	Years mTBI FW 6 (2) mTBI (7) (1) TBI 11 (2)	mTBI FW 45 (2) mTBI 52 (1) TBI 42 (4) HC 42 (1)	mTBI FW 6/8 mTBI 7/25 TBI 5/7 HC 16/24	Digit symbol-coding, reading speed, trail making test, digit span, spatial span, verbal fluency. MFS	Reduced processing speed (digit symbol-coding, reading speed, trail making test) correlated with increased mental fatigue (*p* < 0.05).
Johansson & Rönnbäck, 2015 [69]	76 mTBI 45 HC	9 (8) years	mTBI 43 (12) HC 41 (12)	mTBI 30/46 HC 15/30	A five-minute computer test with five repetitions, including task repetitions with simultaneous load on processing speed and working memory. MFS	During the test, the controls became significantly faster, while this was not found for the mTBI group who remained on a similar slower speed (*p* < 0.05).
Möller, Nygren de Boussard, Oldenburg, & Bartfai, 2014 [64]	24 mTBI 31 HC		mTBI 36 HC 37	mTBI 12/12 HC 13/18	Digit Symbol Substitution Test Executive functions (Color Word Test) Attention (Ruff) FSS	Decreased performance (fatigability) was reported for the mTBI group who reported more fatigue than controls in more demanding cognitive tests with simultaneous activation of several cognitive domains and executive functions (*p* < 0.01). Fatigability was not found to be present for less demanding tests involving simple automatic attention.
Möller et al., 2017 [26]	10 mTBI 10 HC	At least 6 months after	mTBI 38(11) HC 37 (11)	mTBI 5/5 HC 5/5	Psychomotor vigilance task FSS	Reaction time was longer and increased during a 20-min vigilance task for mTBI group, while the reaction time remained stable for the controls. The results showed that RT for patients and controls differed significantly (*p* = 0.005).
Möller, Johansson, Matuseviciene, Pansell, & Nygren Deboussard, 2019 [67]	15 mTBI 15 OC	7–10 days	mTBI 25 (6) OC 28 (7)	mTBI 7/8 OC 11/4	Saccade function Digit Symbol Substitution Test (DSST) The Ruff 2 & 7 Selective Attention Test FSS RPQ was used for self-rated change in fatigue pre- post injury	mTBI scored higher fatigue on RPQ compared with controls (*p* = 0.023) and cognitive fatigability (DSST, *p* = 0.024). FSS did not differ significantly between patients and OC. Acquired fatigue correlated to slower prosaccade performance (RPQ *p* = 0.006). FSS correlated to slower and unstable antisaccade performance (*p* = 0.019). The more fatigability the more errors on the Ruff 2 & 7 subtask-controlled attention (*p* = 0.022).
Rau et al., 2017 [70]	17 mTBI 17 HC	9 (3) months	mTBI 30 (5) HC 31 (5)	mTBI 10/7 HC 8/9	Repetition (3 times) of a working memory and wordlist memory test No fatigue measure, but indirectly suffered from post-concussion.	No interaction for group by time, but significant main effects for time (*p* = 0.04) and group (*p* < 0.001). The mTBI performed worse from time points 1 to 3 (*p* = 0.02). Time 3 showed significant difference between the groups (*p* < 0.001).
Skau et al., 2019 [24]	20 mTBI 20 HC	28 (21) months	mTBI 42 (10) HC 39 (11)	mTBI 7/13 HC 8/12	2.5-h test session One repetition Stroop–Simon Digit symbol coding, Digit span, Symbol search MFS VAS-energy	A significant reduced mental energy for the mTBI after 2.5 h (*p* < 0.01). The controls improved the second time on Digit Symbol Coding (*p* < 0.05), while the mTBI remained on a similar lower level.
Ziino & Ponsford, 2006a [62]	49 mild to severe TBI 46 HC	240 (222) days	TBI 35 (13) HC 34 (10)	TBI 63% males HC 61% males	Complex selective attention task FSS VAS-fatigue	Slower performance on a C-SAT for TBI group comparison with controls (*p* < 0.001), and it was associated with fatigue for mild to severe TBI (*p* < 0.05).
Ziino & Ponsford, 2006b [63]	46 mild to severe TBI 46 HC	240 (222) days	TBI 35 (13) HC 34 (10)		A vigilance test with a duration of 45 min FSS VAS-fatigue	The TBI group was slower and remained at a similar slower speed throughout the duration of the task (*p* < 0.001). Higher state-fatigue was associated with more omissions for TBI. A subgroup of TBI showed a decline in performance (*p* < 0.05).

Abbreviations: sd (standard deviation), HC (healthy controls), OC (orthopedic controls), SDMT (Symbol Digit Modality Test), BNI (Barrow Neurological Institute), MFI (Multidimensional Fatigue Inventory), FSS (Fatigue Severity Scale), MFS (Mental Fatigue Scale), VAS (Visual Analogue Scale), RPQ (Rivermead Post-Concussion Symptoms Questionnaire), mTBI (mild Traumatic Brain Injury).

**Table 2 ijerph-18-05955-t002:** Research relating fatigue in adults after TBI to brain imaging techniques. Summary of study characteristics, including details on study sample, methods, and results.

Reference	Number of Participants Injury Severity	Time Since Injury	Age Year (sd)	Sex (Males/ Females)	Brain Imaging Fatigue Measure	Results
Berginström et al., 2018 [25]	57; TBI 40 mild 10 moderate 7 severe 27 HC	9 (7) years	TBI 42 (13) HC 38 (12)	TBI 31/26 HC 14/13	fMRI, modified SDMT FSS MFS VAS-fatigue	Lower brain activity (fMRI) in basal ganglia, primarily the caudate nucleus, thalamus, and anterior insula for the TBI group compared to controls (all *p* < 0.05). The brain activity decreased across the 27-min test session for the controls, whereas the TBI group remained on a similar lower activity level. Increased state-fatigue after the session compared to controls was reported (*p* < 0.01).
Berginström, Nordström, Nyberg, & Nordström, 2020 [76]	59 TBI; 40 mild 11 moderate 8 severe 27 HC	TBI 42 (9) HC 38 (12)	9 (7) years	TBI 32/27 HC 14/13	WM hyperintensity lesions FSS MFS	WMH lesions were more common in TBI compared to controls. WMH lesions were not related to cognitive tests. Increased WMH lesions correlated with reduced fatigue (*p* = 0.026).
Clark et al., 2017 [78]	59 TBI mild-moderate HC 25	64 (34) months	TBI 33 (6) HC 34 (8)	88% males 72% males	DTI MFIS	Decreased white matter microstructural integrity of left anterior internal capsule (*p* = 0.02) involved in basal ganglia circuitry in mTBI compared to HC, and this was associated with greater level of fatigue (*p* = 0.01).
Clark et al., 2018 [79]	63 mTBI	64 (43) months	32 (6)	87% males	MRI Thalamic volume MFIS	Greater levels of fatigue were associated with decreased right (*p* = 0.026) and left (*p* = 0.046) thalamic volumes. Regional morphometry analysis showed that fatigue was associated with reduction in the anterior and dorsomedial right thalamic body (*p* < 0.05).
Engström Nordin et al., 2016 [29]	10 mTBI 10 HC	At least 6 months Median 5 years	mTBI 37.5 (11) HC 36.9 (11)	mTBI 5/5 HC 5/5	Psychomotor vigilance task (PVT) adapted to MRI FSS VAS-fatigue	Functional connectivity was influenced by PVT task with a significant difference between mTBI and HC (*p* < 0.05) in thalamus and middle frontal cortex, indicating that HC have more extensive functional connectivity network in thalamus and stronger functional connectivity in medial frontal cortex both before and after the PVT task.
Hattori et al., 2009 [74]	15 mTBI 15 HC	At least 6 moths	mTBI 45 (11) HC 43 (9)	mTBI 3/12 HC 3/12	rCBF SPECT PASAT (information processing and sustained and divided attention) Persistent cognitive complaints, specifically the complaint of cognitive fatigue	In all 4 trials, mild TBI had lower PASAT scores (*p* < 0.05). A different pattern of rCBF between mTBI and HC (*p* < 0.05). Less activation for mTBI in the cerebellum and more activation in the prefrontal cortex. mTBI showed dynamic changes in supratentorial rCBF during PASAT, with larger areas of activation bilaterally in the dorsolateral prefrontal cortex and larger areas of suppression in the occipital and parietal cortices.
Kohl, Wylie, Genova, Hillary, & DeLuca, 2009 [27]	11 moderate to sever TBI 11 HC	9 (9) years	TBI 39 (14) HC 38 (9)	More men in TBI compared to HC	fMRI modified SDMT No fatigue scale	TBI group increased activity in the middle frontal gyrus, superior parietal cortex, basal ganglia, and anterior cingulate. Controls decreased activity over time (*p* < 0.005) TBI had slower reaction during all 3 times (*p* < 0.05).
Liu et al., 2016 [72]	25 mTBI acute phase 21 mTBI chronic phase 20 HC	acute; within 2 weeks chronic; more than 12 months	mTBI acute 36 (10) mTBI chronic 36 (13) HC 32 (8)	Acute 15/10 Chronic 12/9 HC 12(8	ASL-fMRI Psychomotor vigilance task (PVT) MFS	PVT was related to arterial spin labeling–fMRI. Sustained attention was impaired in mTBI patients both in acute and in chronic phases comapred to controls, and with worse performance in the acute phase.
Möller et al., 2017 [26]	10 mTBI 10 HC	At least 6 months after	mTBI 38 (11) HC 37 (11)	mTBI 5/5 HC 5/5	rCBF SPEC Psychomotor vigilance task (PVT) FSS	A significant interaction effect between mTBI and HC in several brain regions (*p* < 0.05). In mTBI, at the end of the PVT, fatigability was related to increased rCBF in the right middle frontal gyrus. Self-rated fatigue was related to increased rCBF in left medial frontal and anterior cingulate gyri and decreases of rCBF in a frontal/thalamic network.
Pardini et al., 2010 [80]	97 penetrating brain injuries Divided into; 17 ventromedial prefrontal cortex (vmPFC), 51 dorso/lateral prefrontal cortex (d/lPFC), 29 nonfrontal lesion 37 HC	unknown	vmPFC 59 (1) d/lPFC 58 (0.3) Nonfrontal 58 (0.5) HC 59 (0.6)	All males	CT scan Krupp fatigue scale	Individuals with PBI with vmPFC lesion were significantly more fatigued than the other groups as well as the healthy controls (*p* = 0.013). VmPFC volume correlated with fatigue scores (*p* = 0.0053), the larger the lesion volume, the higher the fatigue scores.
Ramage, Tate, New, Lewis, & Robin, 2019 [73]	60 mTBI 42 OC	At least 60 days prior to assessment 292 (176) days	mTBI 36 (8) OC 33 (10)	mTBI 53/7 OC 40/2	fMRI Functional Connectivity Constant Effort Task FSS	Brain activation associated with effort and fatigue did not differentiate the mTBI and controls, while functional connectivity did. FSS correlated with functional connectivity between the left insula and the dorsal anterior cingulate cortex (*p* < 0.01), the left insula and the right inferior frontal gyrus (*p* < 0.05), and the dorsal anterior cingulate cortex and the right inferior frontal gyrus (*p* < 0.05) medial frontal gyrus correlated with FSS, all during the first half of the 75% effort level.
Schönberger et al., 2017 [75]	53 TBI mild to severe, most moderate or severe 36 subgroup vigilance test	2 (1) years	38 (14)	77% male	MRI total brain volume, and lesions; GM and WM separately as well as combined. Vigilance task FSS	MRI revealed GM and WM brain lesions but fatigue was not related to brain lesions.
Skau, Bunketorp-Käll, Kuhn, & Johansson, 2019 [24]	20 mTBI 20 HC	28 (21) months	mTBI 42 (10) HC 39 (11)	mTBI 7/13 HC 8/12	fNIRS, modified Stroop-Simon, one repetition MFS VAS- energy	Lower event-related oxygenated hemoglobin (oxy-Hb) concentration in the frontal cortex for the mTBI group, compared to controls (*p* < 0.05). No time effect. An interaction (*p* < 0.05) was found, with the mTBI group having a similar lower oxy-Hb concentration for both congruent and incongruent trials, whereas the controls had a higher concentration of oxy-Hb in the more demanding incongruent trial compared to the congruent trial. Higher MFS correlated with lower oxy-HB (*p* < 0.05).
Wylie et al., 2017 [28]	22; 20 TBI moderate to severe, 2 complicated mTBI 20 HC	80 (51) months	TBI 41 (13) HC 38 (11)	TBI 14/8 HC 8/14	fMRI Four blocks of working memory, 2-back task (difficult), and 0-back speed VAS-fatigue	TBI group was slower in response time (*p* < 0.001). Fatigue interacted with task in several areas. Negative correlation between reaction time and fatigue (*p* = 0.08) TBI; correlation between fatigue and activation for 2-back and 0-back were weak (coefficient = 0.0003). HC; there was a positive correlation between fatigue and brain activation for difficult 2-back task (coefficient = 0.0035), and negative for the 0-back task (coefficient = −0.0013). Fatigue in relation to brain activation was related to caudate nucleus.
Wäljas et al., 2014 [77]	48 mTBI 24 HC	27 (9) days	mTBI 36 (12) HC 37 (10)	mTBI 60% females HC 67% females	DTI Post-concussion symptoms (including fatigue but not specifically measured).	mTBI reported more post-concussion symptoms, did not differ on cognitive tests, and had a larger number of low DTI measures (fractional anisotropy values, *p* = 0.003) compared to controls.

Abbreviations: sd (standard deviation), mTBI (mild Traumatic Brain Injury), HC (healthy controls), OC (orthopedic controls), fMRI (functional Magnetic Resonance Imaging), rCBF SPECT (regional Cerebral Blood Flow Single-Photon emission computed tomography), ALS (arterial spin labeling), fNIRS (functional Near-Infrared Spectroscopy), DTI (Diffusion Tensor Imaging), GM (grey matter), WM (white matter), SDMT (Symbol Digit Modality Test), MFI (Multidimensional Fatigue Inventory), FSS (Fatigue Severity Scale), MFS (Mental Fatigue Scale), VAS (Visual Analogue Scale).

## Data Availability

Not applicable.

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
