# Peer review of "Mental Fatigue after Mild Traumatic Brain Injury in Relation to Cognitive Tests and Brain Imaging Methods"

_ijerph, 2021, doi:10.3390/ijerph18115955_

Round 1

Reviewer 1 Report

Dear Author,

The submitted paper was focused on a comprehensive overview of mental fatigue in individuals with traumatic brain injury (TBI). I have understood that individuals with TBI have potentially serious problems regarding long-term fatigue, association between mental fatigue and cognitive function, brain imaging and emotional distress and fatigue in each section of the manuscript. However, there are serious issues regarding contextual constitution of the review article. For instance, there are little information on objective knowledge (subjects demographics, age, gender, onset duration and et al.), statistical information and classification of research design (observational study, intervention study, retrospective or prospective study and et al.). Generally, the review article includes explanation of methodological process for reference retrieval with the flow-chart, the abstract tables according to research design and statistics (e.g. the number of subjects, age, gender, outcomes including fatigue assessment, cognitive items’ score, an amount of oxy-Hb or changes of activation in brain imaging (fNIRS or fMRI) , odds or hazard ratio with 95% confidence interval and et al.). For the above reasons, I kindly suggest that the manuscript needs to be reconstituted.

Author Response

  1. Thank you for your careful review of our manuscript, and the opportunity to submit this revised version for your consideration. Thanks for the recommendations. I have added two tables including type of study and background data for the study as well as results. Table 1 comprising cognitive function in relation to fatigue after TBI, and Table 2 with brain imaging methods in relation to fatigue after TBI. I have also changed the wording and only suggest this to be an overview of the researcher field. I have done search in databases, but according to review standards more than one person needs to be involved.

Reviewer 2 Report

The present manuscript/review is missing from the literature overview for mTBI and fatigue. The author is trying to highlight such a problem. The article is well structured. However, I do miss some neuroimaging studies that have successfully identify mTBI and could be used in the future at the spectrum of thin study. 

Here are the comments

  1. A paragraph for brain imaging techniques used for mTBi identification is missing. The author can include recent studies for mTBI with MEG like Antonakakis et al., 2020, Frontiers in Computational Neuroscience, Antonakakis, et al., 2017, Neuroscience or Vergara et al., 2018, Neuroimage
  2. Temporal Imaging modalities such as EEG and MEG are missing. The author should include a sufficient spectrum of modalities.
  3. In the conclusion section, an outlook for future studies using neuroimaging techniques with machine learning or other kinds of AI to diagnose and/or treat mTBI with fatigue is missing.

Author Response

  • Thank you for your careful review of our manuscript, and the opportunity to submit this revised version for your consideration. I have emphasized that this is about fatigue after TBI in relation to cognition and brain imaging methods. Al lot of studies exist with TBI, mTBI, and brain imaging, but very few have included fatigue. I exemplified for future studies PET and MEG which I really agree with the reviewer as very interesting areas to dig deeper into in relation to fatigue. I did not add AI as this filed with fatigue needs to better understood, but in the future, I agree that AI may be very useful.
  •  I have emphasized that this is about fatigue after TBI in relation to cognition and brain imaging methods. A lot of studies exist with TBI, mTBI, and brain imaging, but very few includingfatigue. See Table 1 and 2.

  • See answer above. I exemplified for future studies PET and MEG which I really agree with the reviewer as very interesting areas to dig deeper into in relation to fatigue. Page 15, line 5-9.

  • I did not add AI as this filed with fatigue needs to better understood, but in the future, I agree that AI may be very useful.

Reviewer 3 Report

In the present paper Johansson provides a review of current literature regarding post-TBI mental fatigue. The review tackles the problem from the cognitive and imaging point of view, giving a comprehensive overview of the current trends and approaches to measure, quantify and treatments. There are some major concerns:

  • Sections 1.4 and 1.5 should be fused, as they are very similar.
  • Emotional distress section should precede imaging section, to clearly separate cognitive and technological sections.
  • The paper shows a heavy unbalance toward the cognitive sections versus the imaging one. While this can be a reasonable approach, the paper should not state as aim to "provide a comprehensive overview of the fundamental role of mental fatigue in relation to cognitive function and brain imaging methods", as regarding brain imaging do not achieve the precision and depth of analysis achieved in the cognitive sections.
  • Even if the aims can be revised to be more cognitive centered, the brain imaging section should be nonetheless expanded.
  • The author should include in the paper a summary table of the literature reported, highlighting cognitive domains discussed in which papers and imaging methods discussed in which papers.

Author Response

Thank you for your careful review of our manuscript, and the opportunity to submit this revised version for your consideration.

  • Sections 1.4 and 1.5 is now fused.
  • Emotional distress section is moved and is now placed before cognitive and imaging sections.
  • The imaging section is expended, with some more references with imagine studies. However, the field when including fatigue is not huge.
  • Two tables are now included. Table 1 comprising cognitive function in relation to fatigue after TBI, and Table 2 with brain imaging methods in relation to fatigue after TBI.

Round 2

Reviewer 1 Report

Dear author,

The author reviews comprehensively about mental fatigue in persons with traumatic brain injury (TBI). As indicated in my review toward the first draft, the information on methodological process, research design and statistics for each reviewed research remains unclear and the author doesn’t resolve these serious issues in resubmitted manuscript. Especially, as the primary process for typical review’s articles, the author doesn’t clearly document the methodology to retrieve references from some databases (Cochrane database, PubMed, Web of Science et al.) using the flow-chart. Because so, I strongly think that the author can’t warrant a quality of articles selected in the current review. In addition, the author doesn’t include statistical information (mean or median [SD or IQR] of outcomes, p-value with 95%CI or odds or hazard ratio in epidemiological studies for each study into the section of “Results” in Table 1 or Table 2, and only the information on demographics (number of participants, duration since injury, age, sex and a name of each fatigue measure) remains insufficiency as well as the firstly submitted manuscript. Reconsidering the these concerns carefully, I would like to kindly suggest that the resubmitted manuscript can be unacceptable for publication of “International Journal of Environmental Research and Public Health (IJERPH)”.

Author Response

Thanks for the comments. This to be an overview of the researcher field. I have included information about the search of articles (last part in section 1.0, page 2, in green). I have also included statistical results (p-values) in the tables – all new is green. I have also added 2 more articles, page 11, in green (Pardini et al and Liu et al), marked with green colour.

Reviewer 3 Report

The authors have adequately addressed all concerns

Author Response

Thanks for the comments.